# Single-protein detection in crowded molecular environments in cryo-EM images

J Peter Rickgauer[1], Nikolaus Grigorieff[1], Winfried Denk[1,2]*

[1]Howard Hughes Medical Institute, Ashburn, United States; [2]Department of Electrons - Photons - Neurons, Max Planck Institute of Neurobiology, Martinsried, Germany

**Abstract** We present an approach to study macromolecular assemblies by detecting component proteins' characteristic high-resolution projection patterns, calculated from their known 3D structures, in single electron cryo-micrographs. Our method detects single apoferritin molecules in vitreous ice with high specificity and determines their orientation and location precisely. Simulations show that high spatial-frequency information and—in the presence of protein background—a whitening filter are essential for optimal detection, in particular for images taken far from focus. Experimentally, we could detect small viral RNA polymerase molecules, distributed randomly among binding locations, inside rotavirus particles. Based on the currently attainable image quality, we estimate a threshold for detection that is 150 kDa in ice and 300 kDa in 100 nm thick samples of dense biological material.

*For correspondence: winfried.denk@neuro.mpg.de

**Competing interests:** The authors declare that no competing interests exist.

## Introduction

Cells depend on specific interactions between their molecular components. Because protein-protein interactions require that proteins be in close proximity, and often require binding in specific configurations, it could be of great benefit to know, for a particular cell, where all its proteins were at one point in time and how they were oriented. Given sufficient precision, this information would not only tell us which proteins are interacting (or, conversely, which are unlikely to do so), but would also allow us to understand better how higher-order, cooperative modes of molecular assembly lead to specific cellular activities and functional states. Much could be inferred from atomic-resolution snapshots of cells, although this is known to be impossible due to radiation damage (*Breedlove and Trammell, 1970*; *Henderson, 1995*; *Glaeser, 2008*). What is possible, however, is the detection of unlabeled proteins in cells on the basis of template matching.

Template matching applied to electron cryo-microscopy (cryo-EM)-based tomograms aims to determine whether certain proteins are present in experimental images by comparing computed protein templates, derived from solved atomic models, to 3D reconstructions of samples. This approach has made it possible to map the 3D locations and orientations of individual ribosomes (*Ortiz et al., 2006*; *Pfeffer et al., 2015*) and proteasomes (*Asano et al., 2015*) in thin regions of unstained frozen-hydrated cells. The optical resolution of a tomogram (~4 nm; *Oikonomou and Jensen, 2016*) allows the 3D locations of individual structures, once detected, to be determined accurately. Detectability is limited, however, to structures large enough to be distinct in shape at this resolution; in practice, only a few protein complexes in the megadalton range have been detected reliably using information in tomograms alone (but see *Beck et al., 2009*; *Kühner et al., 2009*). Template matching in tomograms is difficult in crowded regions of cells because nearby macromolecules can obscure the protein-of-interest's outline and thus interfere with detection

(*Frangakis et al., 2002*; *Grünewald et al., 2003*; *Beck et al., 2009*; *Asano et al., 2016*). Use of a protein's fine internal structure for template matching should allow more robust detection, but also requires higher resolution.

Here we introduce an approach to determine the locations and orientations of proteins in crowded environments by using the high-resolution information that is now available in single cryo-EM images. The approach is based on the fact that–at a sufficiently high spatial resolution–the projected (2D) potential of a protein, and thus the pattern of phase shifts imparted on the electron wave as it passes through the sample, is unique for a particular protein and projection direction. Projection-matching concepts have been used in single-particle analysis (SPA) to select (*Penczek et al., 1992*; *Huang and Penczek, 2004*) and align (*Grigorieff, 2007*) particle images, although low-resolution templates are preferred in order to limit bias in any resulting reconstructions (*Henderson, 2013*). Different from the SPA case, which aims to determine new high-resolution structures, we are here concerned with using existing high-resolution structural information for the unambiguous detection of single proteins, which is not affected by template bias.

We find that effects of macromolecular background are greatly reduced by using close-to-focus images and matched filtering, both of which suppress low-frequency information. We show experimentally that we can reliably detect proteins down to a size limit of 150 kDa in vitreous ice. Simulations that include a macromolecular background suggest that this limit should increase to 300 kDa in a 100 nm thick tissue slice. Even better detectability might be achieved using higher-resolution reference structures. Experiments involving viral proteins in intact rotavirus particles demonstrate detectability in the presence of protein and nucleic acid background.

## Results

### Detection of proteins in isolation

One way to determine whether there is a (target) protein of known identity and orientation at a particular location in an image is to compare the spatial distribution of detected electrons around that location to what is expected for the protein in that orientation and at that location. The expectation value for the electron count in each pixel can be calculated directly from the known three-dimensional arrangement of atoms in the target and the optical parameters of the electron microscope (*Rullgard et al., 2011*; *Vulović et al., 2013*). By suitably varying the projection direction we can generate a set of two-dimensional templates that represent all possible electron distributions (for a given resolution) that this protein could produce in such an image (*Figure 1—figure supplement 1*). Whether an image is likely to contain the target can be established by cross-correlating the image with that template. Cross-correlation allows a match to be detected even when multiple proteins overlap in projection as long as the image-intensity modulation is approximately equal to the sum of the individual proteins' contributions, as is the case for weak-phase objects (e.g. ref. *Reimer and Kohl, 2008*, p. 316). Cross-correlation is able to extract matching information even if individual pixels are dominated by noise as long as the spatial resolution of the microscope as such is preserved, which is the case when using motion-corrected electron movies (*Brilot et al., 2012*; *Li et al., 2013a*).

Searching a single cryo-EM image for a target protein with a high-resolution reference structure requires cross-correlating millions of templates (see Materials and methods, *Figure 1—figure supplement 1*), each corresponding to a different orientation of the target with an image (typically megapixel-sized), or in other words, searching a space of location-orientation combinations (LOCs) with more than $10^{12}$ entries. Since the computational cost of a search goes up linearly with the number of templates, we chose a set of orientations spaced as evenly as possible, thereby minimizing, for a given number of orientations, the largest rotation that is necessary to bring any orientation into alignment with a member of the set. Unlike in Euclidean space, where a regular grid pattern with discrete translational symmetry fulfills this criterion, no simple solution for this problem exists in orientation space (*Saff and Kuijlaars, 1997*) and, after considering several approaches, we settled on a method based on the Hopf fibration (*Yershova et al., 2010*) (Materials and methods). In a set of $\approx 2.4 \times 10^6$ orientations generated in this way, the incremental rotation angle between one set member and its closest neighbor was 1.88 degrees. Out of 10,000 random orientations, none

required a rotation of more than 1.51 degrees to bring it into alignment with one of the members of the Hopf set (*Figure 1—figure supplement 2*).

After confirming in simulated images that correlation-based projection-matching can correctly identify proteins and determine their locations and orientations (data not shown), we tested whether this could also be done in experimental images. We acquired pairs of images of apoferritin (PDB: 2W0O) embedded in vitreous ice, the first image acquired close to the Gaussian focus, where we expected the best detectability given the optical properties of the microscope (see below), and the second image at a more typical underfocus of 2000 nm or more, which readily allowed us to recognize likely target molecules by eye.

For images taken close to focus, where the target molecules are not detectable by eye (*Figure 1a*), there were typically only a small number of orientations for which the corresponding cross-correlograms (CCGs) contained any (and when they did, only a few) pixels with values of the SNR (the ratio of the peak height and the standard deviation of the CCG noise [*Saxton and Frank, 1976*], Materials and methods) that were exceptionally high, i.e., substantially exceeded those expected for a Gaussian noise distribution (*Figure 1b*). High-value CCG pixels were typically clustered at locations that—in high-underfocus images—were encircled by the dark ring typical for apoferritin (*Figure 1d*). The angular differences within the set of orientations associated with a cluster of high-valued CCG pixels were less than three degrees away from the best-detected orientation after taking into account the octahedral symmetry of apoferritin (97.9% of values both >7.32 and within one pixel of any peak, N = 17 peaks).

The sharp dependence of the CCG value on orientation suggested that our search procedure is highly specific to a particular protein structure. To confirm this we searched the image for a decoy protein, GroEL (PDB: 1GRL), of comparable size and molecular weight but otherwise unrelated to apoferritin (*Figure 1c,d*). For GroEL the amplitude distribution of CCG values lacked the high-amplitude tail seen for the target, and instead followed the distribution expected when cross-correlating a set of Gaussian-noise templates with an image also containing Gaussian white noise (*Figure 1e*).

## Radiation damage

Without radiation damage, the SNR should increase with the square root of the accumulated electron exposure because both the peak height and the noise variance increase linearly with the exposure. However, electrons that pass through the sample also destroy structural information, at first mostly at high spatial frequencies but eventually across most of the spectrum (*Glaeser and Taylor, 1978*; *Glaeser, 2008*; *Grant and Grigorieff, 2015*). In our experiments, we found that the SNR (averaged over the ten largest distinct peaks in one image) initially increased at approximately the expected rate (with an exponent of 0.508 ± 0.033), but levelled off at an exposure of around 1000 electrons per $nm^2$ (*Figure 2a*).

## Molecular-weight dependence

Next we explored how the molecular weight (MW) of the target affects the SNR, which we expected to increase with the square root of the MW since the signals due to different parts of the target structure add coherently (and thus linearly), while the noise should add incoherently and thus grow with the square root. To test this, we created a number of model structures that contained only a subset of the atoms in the original apoferritin structure. For each of these fragments, which ranged in molecular weight from 50 to 200 kDa, we calculated templates using the orientations that had provided optimal matches for the whole protein and the other 23 orientations linked by octahedral symmetry. Then, cross-correlating the fragment templates with the experimental images, we found that the average SNR varied with the MW as expected (*Figure 2b*, fitting y = a*(MW)$^\alpha$ to the CCG values for the 5 × 24 fragments and the full template yielded $\alpha$ = 0.508 ± 0.009 and $\alpha$ = 0.514 ± 0.003 for experimental and simulated images, respectively; two particles). This suggested that we should be able to detect proteins down to a MW of roughly 150 kDa when using an SNR of 7.32 as the detection threshold, which should generate about one false alarm (CCG values above the threshold due to noise) per image of 1850$^2$ pixels (Materials and methods). To confirm this, we performed full searches (using our standard set of orientations) for three of the fragments (200, 150, and 100 kDa). We found, respectively, 259, 58, and 10 CCG values >7.32, of which most

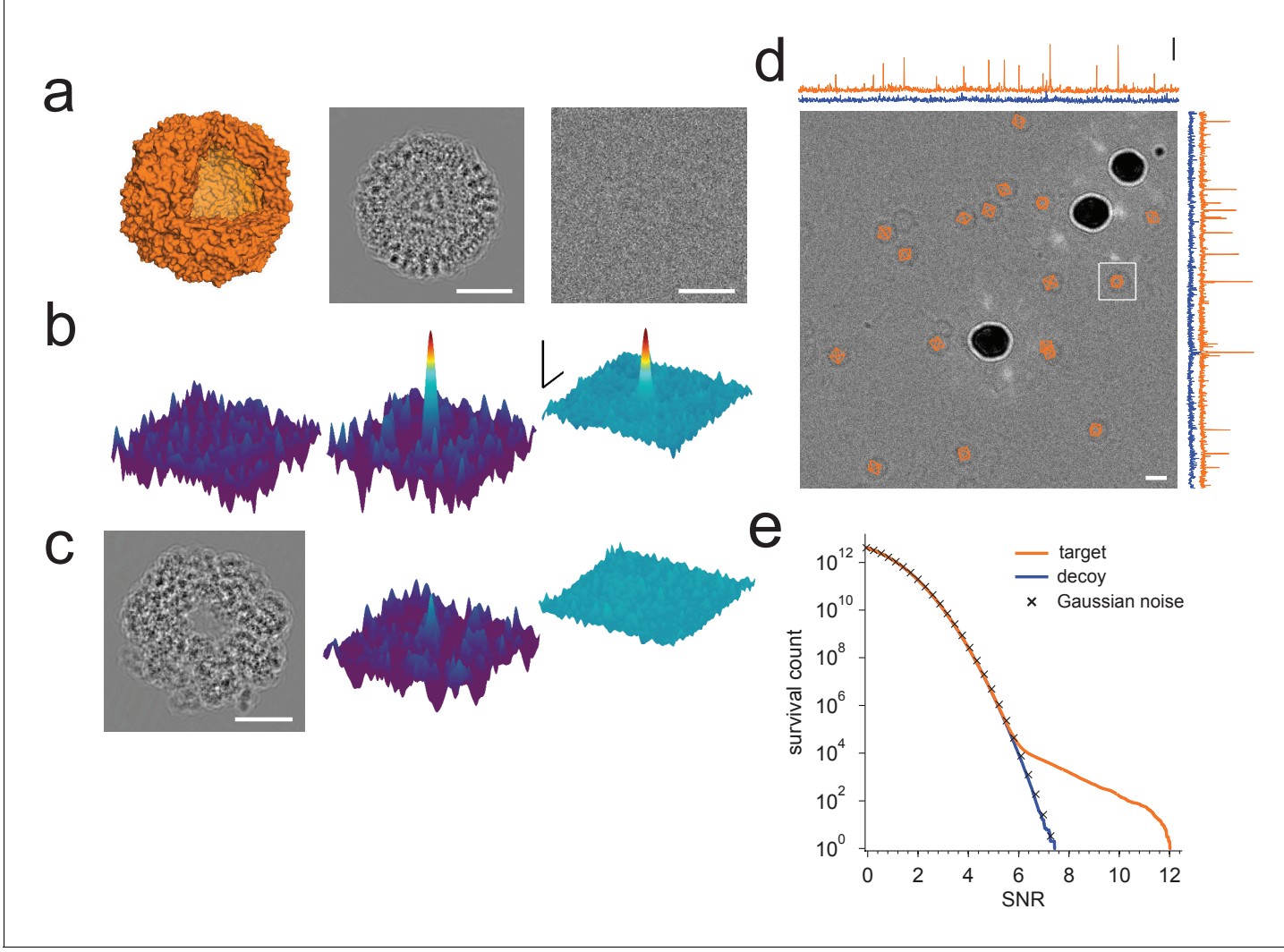

**Figure 1.** Protein detection in vitreous ice. (a) From the left: apoferritin structure, template at 220 nm underfocus, image of a single apoferritin in ice at 220 nm underfocus with 1200 electrons/nm$^2$. (b) Cross-correlograms (CCGs), left to right: template five degrees rotated around z-axis from best orientation, template at best orientation, maximum intensity projection (MIP) across all template orientations. (c) GroEL (decoy) template at 220 nm underfocus, single CCG, and MIP. (d) Image at 2200 nm underfocus. Orange octahedrons indicate the orientation and location of CCG peak values from a full search of an image of the same area at 220 nm underfocus taken before the displayed image. Traces along right and top edge: horizontal and vertical projections of the maximum across orientations (orange: apoferritin, blue: GroEL; blue traces offset by 1.5 SNR units for clarity). Note the dark rings, which presumably correspond to apoferritin particles (the large round objects are gold particles). The boxed region indicates the image region used for (a), (b), and (c). (e) CCG value survival histograms (number of CCG values above a given SNR) for apoferritin (orange), GroEL (blue), and as expected for Gaussian noise (crosses). Scale bars are 5 nm for images and 1 nm for surface plots in (a)-(c), 5 SNR units for surface plots in (b), and 10 nm and 3 SNR units in (d). The top and bottom ends of the amplitude scale bar in (d) correspond to SNRs of 10 and 13, respectively.

The following figure supplements are available for figure 1:

**Figure supplement 1.** The process.

**Figure supplement 2.** Distribution of residual orientation mismatches for a test set of 10,000 random orientations.

(256, 56, and 8) were within 0.3 nm of locations where peaks had been detected using the full-protein templates (440 kDa; *Figure 2—figure supplement 1*).

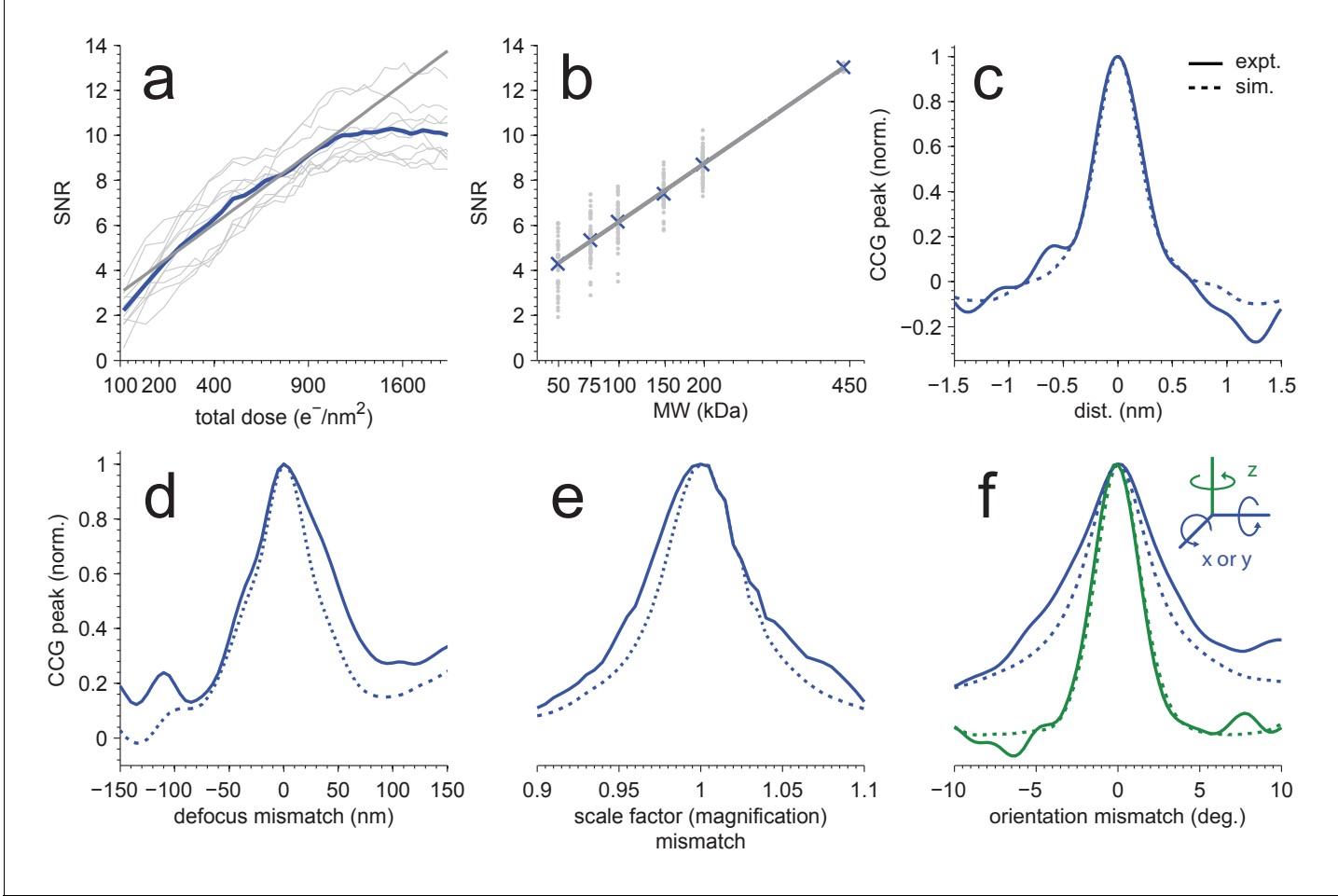

**Figure 2.** Detection sensitivity. (a,b) CCG values at the correct location and orientation *vs.* electron exposure for the particle's full structure (a), and *vs.* the MW (b) for partial structures. The ten (panel a) or two (panel b) particles with the largest SNRs in *Figure 1(d)* were used. Individual and averaged values are shown in gray and blue, respectively. Gray lines show fits to the averages (see Results). (c–f) Peak correlogram values *vs.* lateral position, focus mismatch, scale factor (magnification) mismatch, and orientation mismatch, all normalized to the maximum. Dashed lines are from simulations, solid lines from experiments. Traces in (c–f) are averages of two particles.

The following figure supplements are available for figure 2:

**Figure supplement 1.** Detection using template fragments.

**Figure supplement 2.** Sensitivity to template errors.

## Mismatch sensitivity

Some or all target proteins in an image might remain undetected if the wrong microscope parameters are assumed or if the search-space is undersampled with respect to orientation or position. To determine how well we need to know the experimental defocus and magnification, and how densely we need to sample location and orientation, we mapped for both experimental and simulated images how the peak height in the CCG varied with these parameters. We found (*Figure 2c–e*) that a reduction in the SNR by 20% was caused by a positional misalignment of 0.15 nm in experimental images (0.13 nm in simulations), a focus mismatch of 21.0 nm (15.5 nm), or a magnification error of 1.9% (1.6%). The sensitivity to errors in template orientation depends on the rotation axis. While for the axis perpendicular to the image plane (z-axis) a mismatch of 0.95 degrees (0.89 in simulations) was sufficient for a 20% drop in SNR, for an axis parallel to the image plane 1.92 degrees (1.34) were needed (*Figure 2f*). The lower sensitivity for rotations around an in-plane axis is likely caused

by the foreshortening of the atomic movements as their z-components are lost upon projection. Using simulations, we also explored the effect of the focus setting on detection and found that at 2000 nm underfocus the peaks in the CCG flare out towards a broad base but still possess a sharply defined maximum (*Figure 3a*) with a peak SNR quite comparable to that seen for much smaller defocus values. This means that, at least for isolated target proteins, the detectability did not depend strongly on the defocus.

## Protein background (simulations)

How is detection affected by other objects in the sample, such as proteins located above or below the target? While in brain tissue only about 10% of the mass density is due to lipid and protein (*McIlwain and Bachelard, 1985*) with the remainder being mostly water, it is likely that other proteins will interfere with the detection of a target protein much more strongly than water at the same projected mass density because proteins resemble each other more than they resemble vitreous water. We explored this situation by simulating, at 70 nm and 2000 nm underfocus, images of a synthetic sample that contained apoferritin as the target protein, and a background of randomly oriented and placed BSA proteins at an average density of 37.5 kDa/nm$^2$. The apoferritin target could not be detected by eye in either image, yet searching the 70 nm image yielded a clear peak at the correct location albeit with reduced SNR compared to the background-free case (14.4 *vs.* 16.9). Template matching was, however, no longer able to reliably detect the target in the 2000 nm image (*Figure 3a*, *Figure 3—figure supplement 1*).

We suspected that the reason for this difference is that the defocus setting profoundly affects the contrast transfer function (CTF). As the defocus becomes smaller, low spatial frequencies are increasingly suppressed, the number of phase reversals is reduced, and the suppression of high spatial frequencies, which is caused by the fact that the illumination is only partially coherent, becomes much less severe (*Figure 3b*). To determine which of these effects, if any, are responsible for the observed behavior, we first simulated images as they would be generated by a (hypothetical) phase-plate microscope (PPM) with a CTF that is constant in both amplitude and phase (except for the 90 degree phase shift due to the phase plate everywhere but near zero spatial frequency) and thus attenuates neither high nor low spatial frequencies. In simulated PPM images the target protein was neither visible by eye nor could it be detected by cross-correlation (*Figure 3a*, *Figure 3—figure supplement 1*). This was somewhat surprising because PPM images contain nearly all the phase-contrast information that is available; no information is lost to partial coherence or to phase reversals. This suggested that the loss of detectability against protein background seen for simulated PPM and large-defocus images is caused by low-spatial-frequency noise that happens to be suppressed by the close-to-focus CTF.

For simulations that included protein background, the image noise was dominated by low-frequency components, both at 2000 nm defocus and for the PPM (*Figure 3c*). This suggested that to optimize detection one needs to use the full matched-filtering concept (*McDonough et al., 1995*), which includes, in addition to template matching, a whitening filter that flattens the power spectrum of the image noise, maximizing detectability against a spectrally inhomogeneous background. Our whitening filter comprised the reciprocal square root of the radially averaged power spectral density (PSD; *Figure 3c*), and was applied by Fourier-space multiplication to both the transformed image and template before calculating each cross-correlation (Materials and methods).

We found that whitening, in fact, restored detectability: CCGs now showed strong and narrow peaks at the correct locations (*Figure 4a*). Detectability was best (SNR = 20.5) for the PPM followed by 70 nm (SNR = 14.3) and 2000 nm (SNR = 7.7) underfocus. For the PPM and for 2000 nm underfocus, the whitening filter strongly suppressed low-frequency information (*Figure 4b*). These simulation results confirm that a major reason for compromised detectability in strongly defocused images is that there the CTF, unlike in close-to-focus images, does not suppress the low-frequency noise that dominates images with a high density of macromolecules.

We also confirmed that a substantial part of the difference in SNRs between 70 and 2000 nm underfocus was due to illumination-coherence effects by repeating some of the simulations with completely coherent illumination. This increased the SNR at 2000 nm by 42% (from 7.7 to 10.9), but only by 8% at 70 nm (from 14.3 to 15.4). The remaining difference in SNR is likely due to the fact that near the Scherzer focus (70 nm defocus) the CTF remains very high in a part of the spatial frequency range that provides much information (*Figure 3b*).

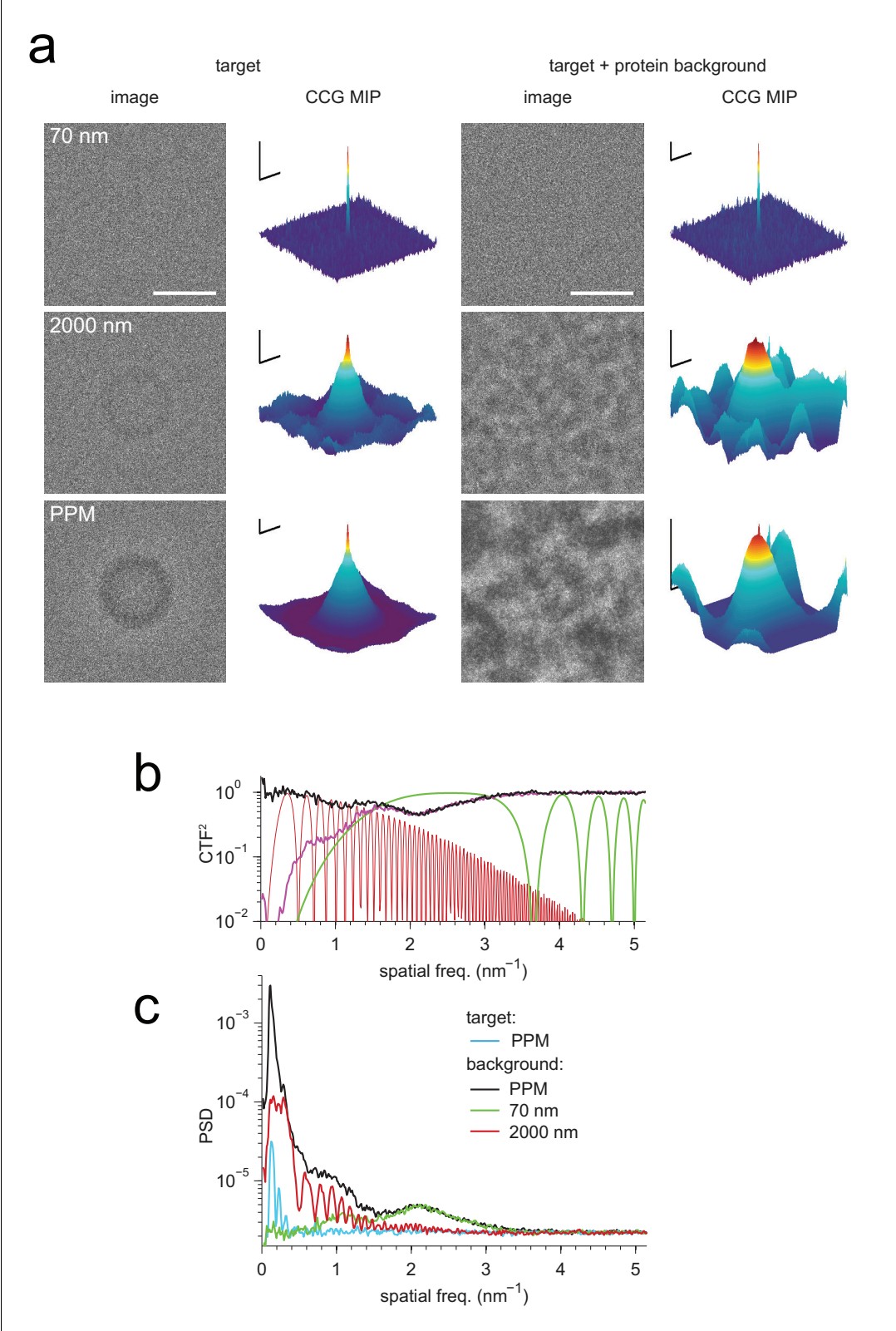

**Figure 3.** Detection against background. (a) Simulated images of a single apoferritin in ice without (left) and with (right) a dense protein background (BSA at 37.5 kDa-nm$^{-2}$), at 70 nm and 2000 nm underfocus as well as for a perfect phase-plate microscope (PPM). To the right of each image the corresponding maximum-projected CCG from a full orientation search is shown. (b) Squared contrast transfer functions (CTFs) for 70 nm (green) and 2000 nm (red) underfocus, and squared whitening filters for 70 nm underfocus (black) and the PPM (purple). (c) Power spectral densities for the

*Figure 3 continued on next page*

*Figure 3 continued*
apoferritin template alone (light blue) and for simulated images of the protein background alone using the PPM (black), at 70 nm underfocus (green), and at 2000 nm underfocus (red). Spatial scale bars are 10 nm for the images and 2 nm for CCGs. SNR bars are 5 and 2 SNR units for left and right columns in (a) respectively.
The following figure supplement is available for figure 3:

**Figure supplement 1.** Full-search CCG MIPs for simulated images of apoferritin with BSA background, 2000 nm underfocus (*Figure 3a*, right column), without (**a**) and with (**b**) pre-whitening.

We also explored the SNR for a whole range of defocus values from 500 nm overfocus to 2000 nm underfocus (*Figure 4b*). Within this range, the best SNR was seen at 70 nm. Against a protein background the SNR varied strongly with defocus near the Scherzer setting and then fell steadily with focus distance (*Figure 4b*), mitigated somewhat by whitening. In contrast, a background comprising randomly placed carbon atoms, which produced a spectrally flat noise distribution (data not shown), reduced protein detectability uniformly for all defocus settings (*Figure 4b*). Why don't coherence effects reduce the SNR at large underfocus without protein background (*Figure 4b*)? One possibility is that a gain in low-frequency information, which is eliminated by the whitening filter in the protein-background case, compensates for the loss of high-frequency information caused by the coherence envelope.

We next determined how much different spatial frequencies contribute to target detectability by removing all spatial frequencies above a variable frequency threshold and observing the effect this has on the SNR (*Figure 5a,b*). For an isolated apoferritin molecule in vitreous ice, the average SNR for five of the particles in the experimental image increased until it reached a value of about 12 at a spatial frequency of 3 nm$^{-1}$ after which it essentially stayed constant. In a simulation using the same parameters the SNR closely followed the experimental curve up to a resolution of 2.4 nm$^{-1}$ beyond which it diverged upward to reach a value of 17.6 at 5.2 nm$^{-1}$. The difference could be due to residual magnification or focus mismatch, uncorrected sample drift, or astigmatism in the experimental data (*Figure 5a*). In simulations with protein background (*Figure 5b*), the SNR for matched-filter detection reached maximum values of 14.7 and 8.3 at defocus settings of 70 and 2000 nm, respectively. The PPM outperformed both settings throughout.

Why does the SNR rise only slowly beyond 3 nm$^{-1}$ even in simulations, where factors such as residual astigmatism, focus mismatch or uncorrected sample motion, all of which suppress high-frequency information, do not play a role? We suspected low-pass filtering of the reference structure to be responsible. When instead of using for each atom the B-factor listed in the PDB entry, which are on average 19.6 Å$^2$ (corresponding to Gaussian low-pass filter with $\sigma = 3.2$ nm$^{-1}$) we used a uniform value of 5 Å$^2$ ($\sigma = 6.3$ nm$^{-1}$), the SNR curves (all without ice) no longer plateaued. The values reached at the highest frequency (5.2 nm$^{-1}$) increased by 37.3% for the PPM and by 33.8% and 19.9% for 70 nm and 2000 nm underfocus, respectively (*Figure 5b*).

What ultimately matters for detectability is the trade-off between precision (the fraction of apparent detections that are correct) and recall (the fraction of targets detected). Using full searches of simulated images of a synthetic sample containing 50 apoferritin proteins at random locations and orientations together with a BSA background, we determined precision-recall curves (PRCs) for a number of different conditions. Again and largely independent independent of the SNR threshold (which is varied to generate the PRC) images close to focus performed best, even when assuming perfect illumination coherence (*Figure 5c*).

## Protein background (experiments)

To test whether single proteins can be detected in actual cryo-EM images of densely protein-packed biological samples, we analyzed images of rotavirus double-layered particles (DLPs), some taken specifically for this study and some from a prior study (*Grant and Grigorieff, 2015*). Rotavirus DLPs are 70 nm in diameter and have at their core a densely-packed double-stranded RNA genome (*Estrozi et al., 2013*). The genome is contained inside a protein shell assembled in perfect icosahedral symmetry from two protein types, the inner capsid protein, VP2 (120 copies, approximately 90 kDa each), and the outer capsid protein, VP6 (780 copies, 45 kDa each), which together form 60

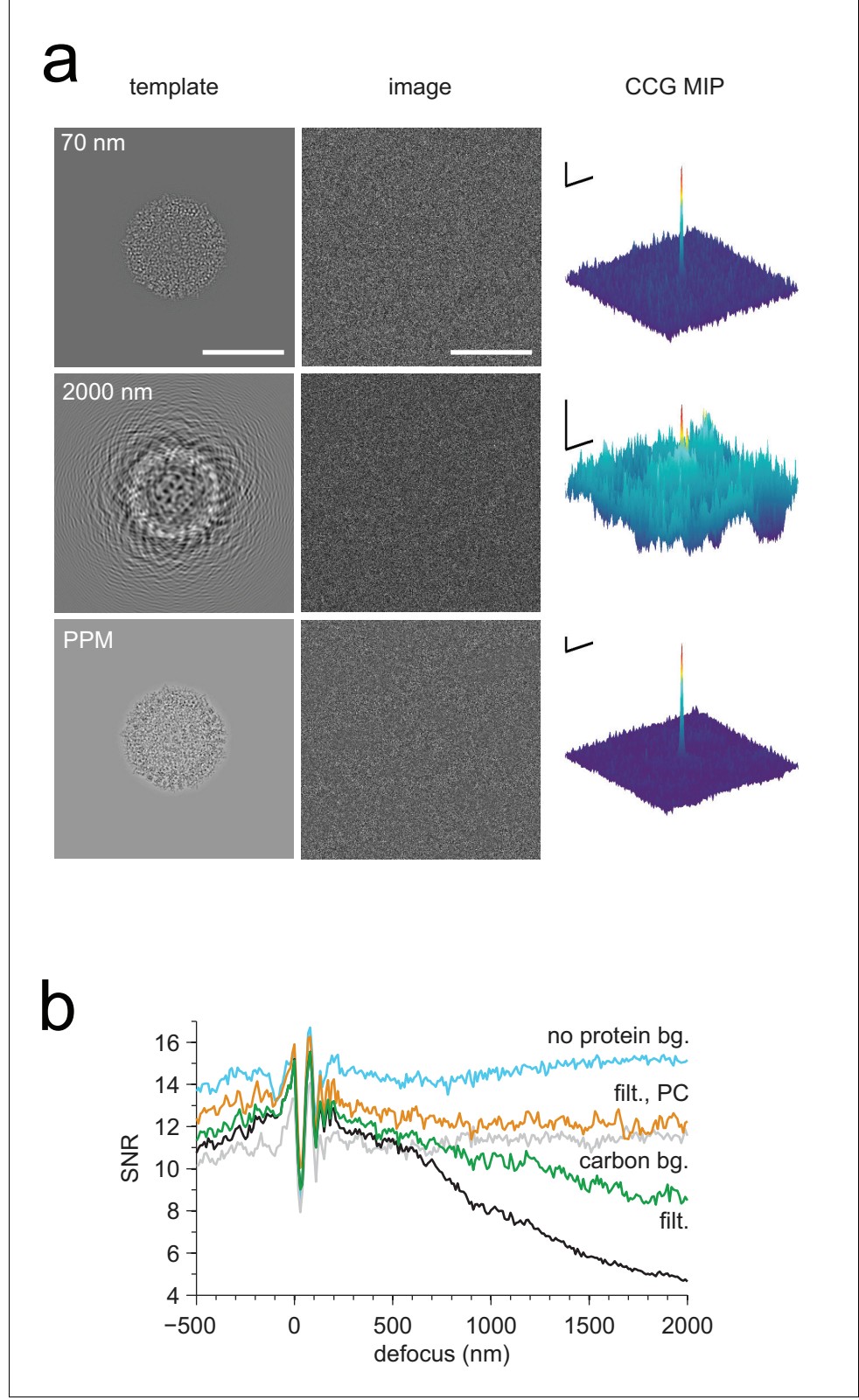

**Figure 4.** Optimized detection. (**a**) Whitened templates and simulated images together with the corresponding maximum-projected CCGs for apoferritin with a BSA background of 37.5 kDa/nm$^2$ (same as in *Figure 3a* right except for whitening). (**b**) SNR *vs*. defocus for simulated images of apoferritin, all with 50 nm of ice but only some with BSA background (black, green, orange), with whitening (green, orange), with randomly scattered carbon

*Figure 4 continued on next page*

*Figure 4 continued*

atoms (18.7 kDa-nm$^{-2}$) as background (gray), and with perfect illumination coherence (orange). Scale bars are 10 nm for images and 2 nm and 2 SNR units for CCGs.

asymmetric subunits (ASUs, 720 kDa, *Figure 6* inset). In addition to these structural proteins, each DLP contains at least two enzymatic proteins, VP1 (at least 11 copies, 115 kDa each) and VP3 (at least 11 copies, 90–100 kDa each), which together are responsible for synthesizing and capping single-stranded RNA molecules before they are extruded into the infected host cell (*Ogden et al., 2014*). The projected density of biological macromolecules in the DLP is around 20 kDa nm$^{-2}$ across most of the particle, which is similar to the density (20–30 kDa nm$^{-2}$) expected for a 100 nm-thick section of tissue (*Ellis, 2001*). Because of the icosahedral symmetry, the orientation and projected location of each of the 60 ASUs can be predicted from the location and orientation of the entire DLP and can thus be used to assess the detection performance.

We first analyzed images from 10 DLPs, five each taken near (270–310 nm) and far (1400–1600 nm, *Figure 6a*) from focus. In high-defocus images, DLPs are easily detectable by eye (*Figure 6a*). When searching these images for ASUs, using the full set of Hopf rotations and a whitening filter, we found that most (93.3% for 1400–1600 nm defocus and 99.7% for 270–310 nm defocus) of the CCG values above the detection threshold occurred within 0.5 nm and five degrees of the expected locations and orientations (*Figure 6b*). The superiority of images taken close to focus, a central prediction of our simulations, persists along most of the precision-recall curve (*Figure 6c*).

## Constrained searches

The number of pixels where the noise alone generates CCG values above a given threshold increases with the number of independent LOCs searched while the threshold needed to obtain a certain detection rate decreases as the molecular weight of the target falls (*Figure 2b*). This means that for proteins below a certain size a meaningful search of the entire LOC space is no longer possible, which puts many proteins out of reach. Our data (*Figure 2b*) suggest that the size limit is around 150 kDa for isolated targets, and should be about twice that with protein background, which reduces the SNR by about 30% (*Figure 4b*). While prior information about the location or orientation of the target does *not* reduce the chance that for a particular LOC the detection threshold is breached by the noise, it does reduce the total number of false alarms because fewer LOCs need to be searched and thus should allow the detection of much smaller targets.

We explored this idea by performing a constrained search for VP1, an RNA-dependent RNA polymerase that binds to the inner surface of the DLP capsid (*Estrozi et al., 2013*). Given that we know the LOCs of all ASUs, the number of LOCs that need to be searched is reduced from about 10$^{12}$ to below 10$^3$. The search was based on a hybrid molecular model (PDB: 4F5X) that combines the structure of VP1, as determined by X-ray crystallography (PDB: 2R7O), and an independently determined structure of the ASU (PDB: 3KZ4, see also Materials and methods). The dataset contained 4,178 DLP images (*Grant and Grigorieff, 2015*), taken at focus settings between 300 and 1900 nm. First, we determined possible LOCs for VP1 by searching with a template set based on only the ASU portion of the hybrid model. We retained only those DLPs (3,296) where we found all 60 ASUs when using a threshold that returned one false alarm per image (750 by 750 pixels). Then, using only the VP1 portion of the model, we searched only those LOCs that agreed exactly with the orientation of one of the ASUs and were within one pixel (0.1 nm) of its location.

Using a threshold of 2.03, which yielded, on average, 0.059 false alarms per ASU LOC for a set of control templates (Materials and methods), we found 9.63 VP1s per DLP. Assuming 11 bound VP1s per DLP, about 2.9 (0.059 × (60-11)) of those should be false alarms leaving 6.73 true positives, a recall of 71.2%. The prevalence of vertices that contained one detected VP1 was substantially larger (and those with zero, two, or three substantially smaller) than the binomially distributed prevalence one would expect for independent binding and what one gets with the binding sites randomly shuffled within each DLP (*Figure 6d*). This is consistent with the assumption that steric hindrances prevent the binding of more than one VP1 per vertex (*Estrozi et al., 2013*).

The SNR of 2.28, estimated from CCG histograms (Materials and methods), is smaller than the value of 2.98 that one gets by extrapolation from 7.45, the mean peak value seen when searching

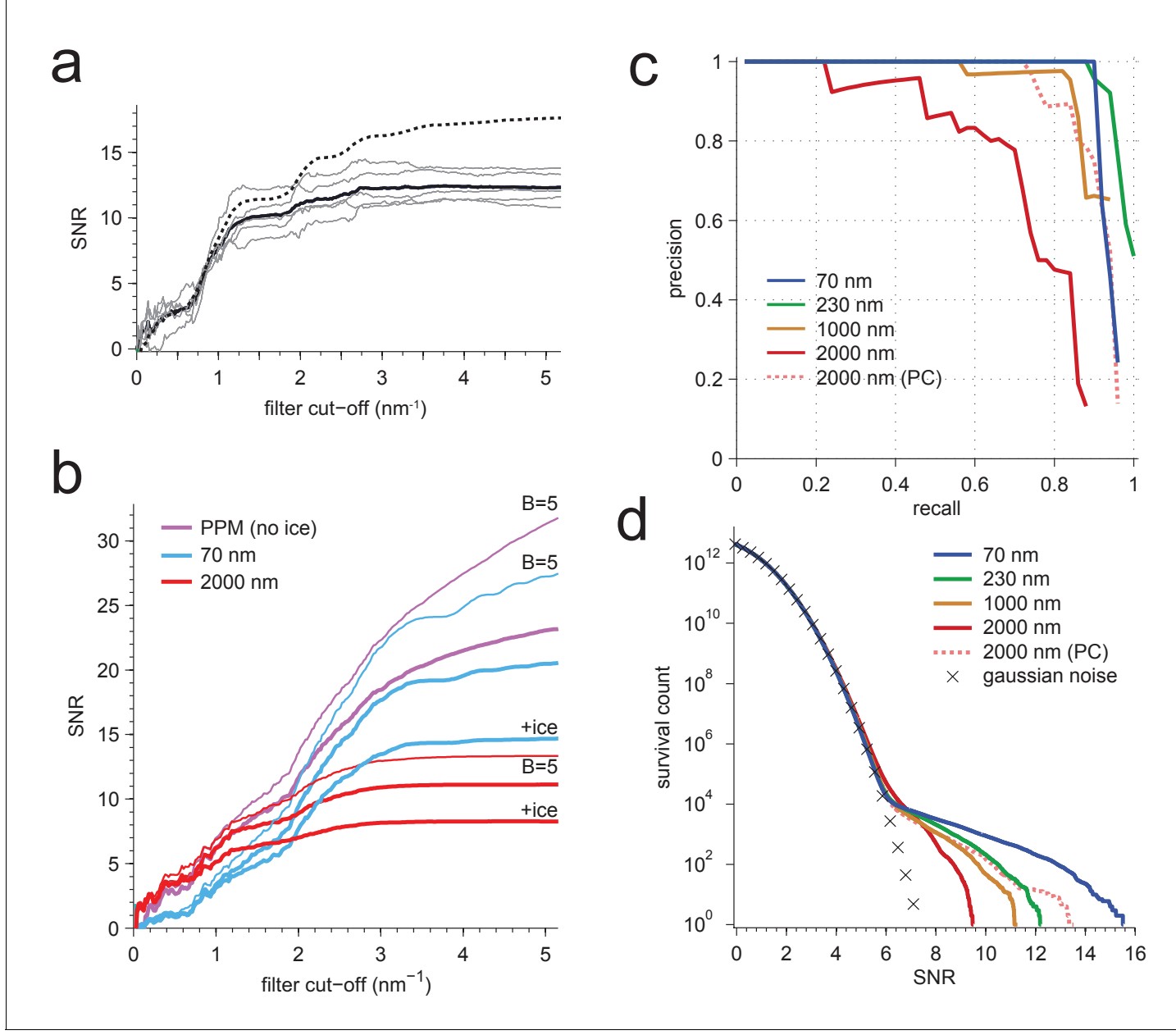

**Figure 5.** Resolution dependence and performance. (a,b) SNR vs. low pass-filter cut-off; (a) for five of the particles in *Figure 1d* (individual trace: thin gray, average: thick black) and the average of two simulations (dashed) using the same optical parameters as for two of the experimental particles; (b) for simulated images using the protein background, whitening and B-factors as in the PDB file (thick traces) or B = 5 Å² (thin traces) and ice or no ice as indicated. (c) Simulated-image precision-recall curves for 50 randomly oriented and positioned apoferritin molecules with 37.5 kDa-nm⁻² of BSA background at underfocus values as indicated. Full orientation searches with whitening and standard imaging parameters were used, except in one case, which used perfect illumination coherence (2000 nm, PC). (d) CCG value survival histograms (number of CCG values above a given SNR) from full searches of simulated images with parameters as indicated. Crosses: Gaussian noise (same as *Figure 1e*).

The following figure supplement is available for figure 5:

**Figure supplement 1.** Survival histograms for various simulated image and template-matched conditions as in *Figure 5d*, additionally showing results for a PPM and a 2000 nm underfocus image that was not whitened (all other traces reflect whitening).

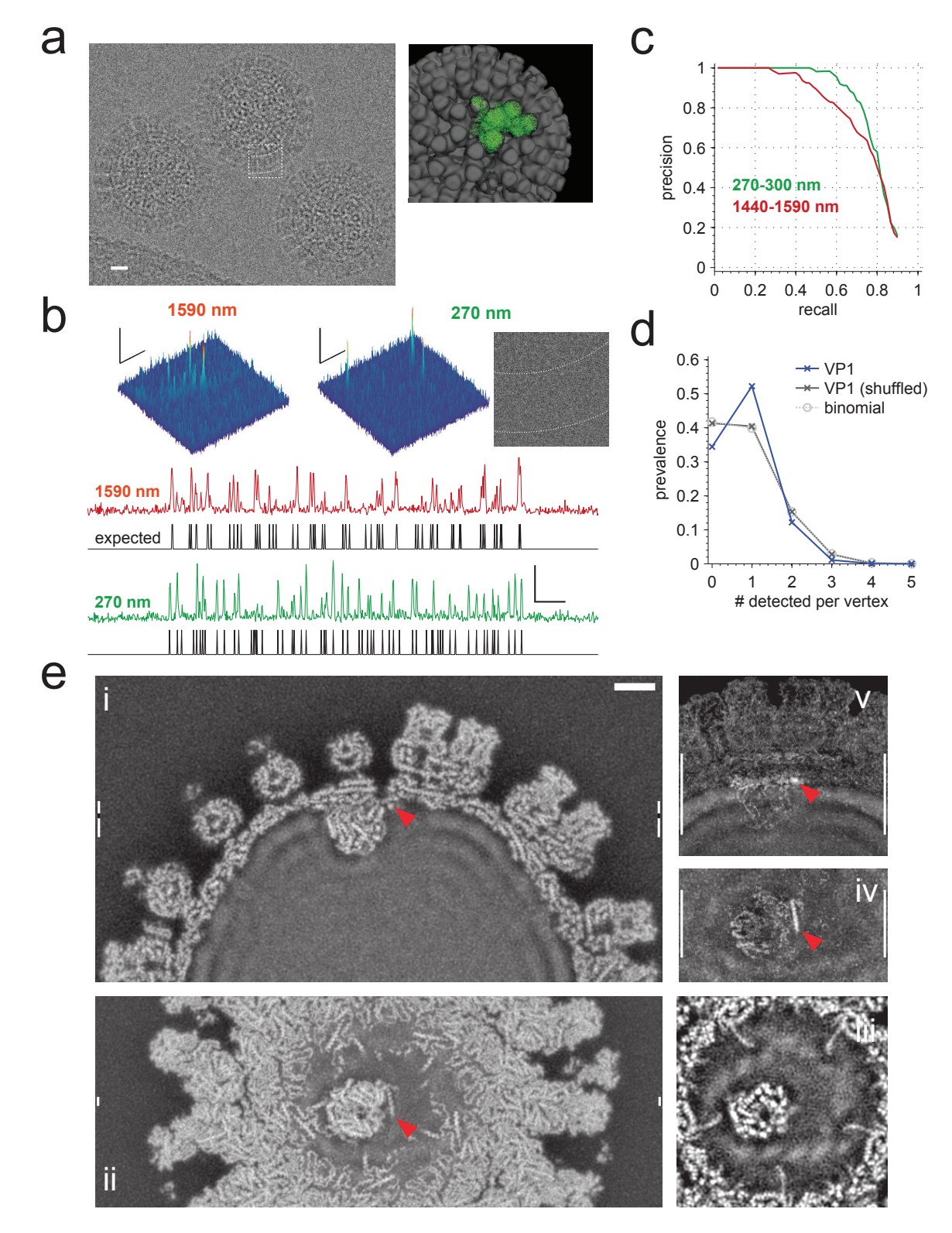

**Figure 6.** Application to rotavirus. (a) Left: rotavirus DLPs imaged at 1590 nm underfocus with 1600 electrons $nm^{-2}$. Right: surface-rendered electron density of the DLP with one ASU highlighted in green. (b) MIP CCGs from searches for the ASU in images taken at 1590 nm and 270 nm underfocus, respectively. Top: regions around an ASU peak and (only for 270 nm) the corresponding image region with inner and outer capsid edges indicated by dashed lines. The corresponding region in the 1590 nm image is indicated by the dashed square in a). Below: maximum projections over all

*Figure 6 continued on next page*

*Figure 6 continued*

orientations and one spatial direction and the expected peak locations (black traces). (**c**) Precision-recall curves for detection of the ASU; five DLPs each acquired with an underfocus between 1440 and 1590 nm (red) and between 270 and 300 nm (green). (**d**) Prevalence of the number of VP1 polymerase proteins per fivefold vertex detected in a constrained search (dark blue) and after randomly permuting vertex labels (gray, crosses) and for a binomial distribution with the same mean detection rate (dashed, circles). (**e**) Density map from a VP1 detection-triggered reconstruction of a rotavirus DLP: i) MIP of a 1 nm thick subset (extent indicated by white vertical bars in (ii). Note the polymerase to the left of the RNA exit channel in the capsid. ii) Orthogonal MIP of a different subset (location indicated by white vertical bars in (i). iii) average over a 0.5 nm thick subset (location indicated by gaps in white vertical bars in (i). Note the presumptive RNA helix wrapped around the polymerase. iv) and v) MIPs of the difference between reconstructed and a simulated potential based on PDB:4F5X (projected ranges given by white vertical lines in (v) and (iv), respectively). Red arrow indicates possible VP3 helix. Scale bars are 10 nm in (**a**), 5 nm and 3 SNR units in (**b**) and 5 nm in (**e**).

for ASUs, using the square-root dependence of the SNR on the molecular mass (*Figure 2b*). This is likely due to differences between the X-ray-derived and the in situ structure of VP1 (see below). Note also that even 2.98 is still substantially smaller than 4.3, the value expected using the SNR of around 6.16 seen for an apoferritin fragment with about the same MW as VP1 (100 kDa, *Figure 2b*) and taking into account the roughly 30% reduction in SNR caused by the macromolecular background (*Figure 4b*).

The ability to detect individual VP1s allowed us to reconstruct, by direct Fourier inversion (*Grigorieff, 1998*), the structure of VP1 bound to one of the five sites at the fivefold vertex. We used for each image only the shifts and orientations of the LOCs at which VP1 molecules, assumed to belong to a single class, were detected (*Grigorieff, 2007*), but did not use further refinement of positions or orientations or assume any symmetry. The resulting potential distribution includes a well-resolved VP1 molecule attached to one of the five possible binding sites near an ssRNA exit portal (*Figure 6e*). Unlike detection, reconstruction can be affected by template bias, but the differences between our VP1 structure and the structure used to generate the templates, which are most numerous near the contact to the capsid (*Figure 6e*, *Supplementary file 1*) and include an $\alpha$-helix in close contact with both the polymerase and the exit portal (*Figure 6e*, red arrow), cannot due to template bias (*Henderson, 2013*) as they are not part of the template. This helix could well be part of the RNA-capping enzyme VP3, whose structure (*Brandmann and Jinek, 2015*; *Ogden et al., 2015*) contains a helix of appropriate length and is proposed to reside near VP1 (*Gridley and Patton, 2014*).

In addition to proteins, the reconstruction also shows structural features in the nucleic acid regions of the density map near VP1 that were not seen in icosahedrally symmetric reconstructions (*Estrozi et al., 2013*) (*Figure 6e*).

## Discussion

We have shown that high-resolution template matching can detect—with high selectivity—proteins of known structure in single cryo-EM images and determine—with high precision—their orientation and projected location. When combined with whitening this approach can detect moderately sized proteins even when they are surrounded by a high density of other proteins, as is the case when trying to detect components of individual macromolecular assemblies and determine their spatial arrangement. While some of the issues that arise are also encountered when determining molecular structures by single-particle averaging, the need to ensure high sensitivity and high selectivity in the presence of a dense background of other biological macromolecules is not one of them. We found that operating close to focus helps in the detection of target proteins first by suppressing low spatial-frequencies (which are critical when picking particles for single-particle averaging) caused by the presence of other proteins and, second, by preserving high frequency information crucial for sensitivity and selectivity. While the suppression of low frequencies can also be achieved by filtering the acquired images, the irreversible loss of high-frequency information that occurs due to partial-coherence effects becomes more severe at larger defocus and was responsible for the degraded detection sensitivity seen in simulated large-defocus images.

While the inclusion of higher spatial frequencies in both templates and in the image improves the detection sensitivity (*Figure 5b*), it also exacerbates the differences between a particular reference structure (usually determined in a non-native context such as a crystal) and its conformation in situ

(*Figure 6e*, *Supplementary file 1*). Searches may, therefore, need to include template sets for a whole range of possible conformations. Finally, magnification and focus will need to be estimated more precisely and the orientation, in particular, sampled more densely.

Without prior knowledge about their location and orientation but otherwise optimal conditions we can detect proteins with a molecular weight above about 150 kDa when suspended in vitrified ice. Whether detection of proteins above 300 kDa is possible in 100 nm-thick slices of vitrified biological material should be testable with the help of modern cryo-sectioning techniques (*Al-Amoudi et al., 2007*).

How does our detection sensitivity compare to the ultimate limit, given by the number of scattered electrons? At 300 keV and an exposure of 1000 electrons/nm$^2$, a single apoferritin molecule scatters about 1436 (Materials and methods) electrons elastically, of which about 20% are available at a resolution of 0.4 nm. Taking into account that due to the oscillation in the CTF only half of those are usable we expect a SNR of $\approx 12$ ($(1436 \times 0.2 \times 0.5)^{0.5}$), which is close to what is seen (*Figure 1*). If all scattered electrons could be used, the SNR would increase to almost 38. The minimal molecular weight—assuming perfect electron-optical resolution, a B-factor of 0, no background, and a SNR of about seven needed for detection—would then be around 16 kDa.

Detection of the rotavirus RNA polymerase and the mapping of the occupied binding sites inside the virus capsid (*Figure 6d–e*) illustrates how one can analyze partially stochastic protein assemblies, in which a ligand protein (here, VP1) binds at only some of the available binding sites on a molecular host (the virus capsid). Such assemblies play an important role, for example, during the establishment and control of synaptic function. Often the structures of key components and their interaction partners are known, but whether in a particular synapse this interaction is utilized may be impossible to determine by conventional methods yet might yield to this type of analysis. If the steric constraints of the interaction are known (*Figure 6d–e*), even ligands below the unconstrained detection limit should be detectable.

## Materials and methods

### Specimen preparation

Apoferritin samples were prepared to a final protein concentration of 0.25 mg/mL in PBS from horse-spleen apoferritin (#A3660, lot #SLBF8335V, Sigma-Aldrich, St. Louis, MO) and contained 15 nm gold particles (#1115, Nanoprobes, Yaphank, NY). A 3 µL aliquot of the solution was applied to a freshly glow-discharged (60 s. at 15 mA, Easiglow, Ted Pella, Inc., Redding, CA) holey carbon support grid (#CF-1.2/1.3–4C, Protochips, Morrisville, NC) and plunge frozen using a Vitrobot Mark two (FEI; 5 s. blotting time, blotting force 5, 85% relative humidity). The sample was then transferred into liquid nitrogen and stored until imaging.

### Microscope alignment and calibration

Electron cryo-microscopy was performed on a transmission electron microscope (TEM) operated at 300 kV (Titan Krios, FEI, Hillsboro, OR) in parallel illumination mode. The objective aperture was removed and the condenser aperture was set to 70 µm, the spot size to 4, and the illuminated area diameter to 1.70 µm. State-of-the-art low-dose procedures were used to limit sample exposure during microscope alignment and low-magnification (40X-1700X) inspection of the sample, which preceded each high-resolution imaging session. Particular attention was paid to positioning the specimen near the microscope's eucentric focus as well as to minimizing astigmatism and coma, using the objective stigmator controls and coma-free alignment tools in the Titan User Interface (FEI).

Reference images to calibrate the direct electron detector (DED) pixel dark current and gain were acquired immediately before each set of experiments with the column closed (for dark current) or by imaging an open square (for gain reference). Exposure rates were estimated for the illumination and detection configurations used for high-resolution image acquisition of each sample (see below) and, when necessary, the spot size and beam intensity settings were changed to keep the rates below five electrons pixel$^{-1}$ s$^{-1}$, in order to limit coincidence losses to around 6% (*Li et al., 2013b*). Microscope and DED were controlled by SerialEM (*Mastronarde, 2005*) and Digital Micrograph (K2 Summit, Gatan, Inc., Pleasanton, CA). Image data were recorded as movies with an exposure time of 25

msec per frame at a nominal magnification of 59,000 in counting mode, which resulted in a sample-referred pixel pitch of 0.0482 nm. The dark current was subtracted at the point of acquisition.

## Data pre-processing

Movies were gain-corrected by multiplying all frames by the gain reference. A pixel location was considered 'hot' if it had a mean value of more than seven times the standard deviation above the mean of all pixels in the movie. Each hot pixel was set to the mean of the eight surrounding pixel values in each frame. Typically, fewer than 20 out of $1.4 \times 10^7$ pixels were hot. Each frame in the movie was then cropped to a centered square, down-sampled by a factor of two using Fourier cropping, and real-space cropped to a final size of $1850 \times 1850$ pixels.

To estimate the motion-correction shifts for each movie frame, the down-sampled movies were low-pass filtered with a cosine-edge mask rolling off between 4.6–4.8 cycles $nm^{-1}$ and then analyzed with a custom algorithm that progressively incorporated movie frames into a registered stack (R) while also iteratively refining the alignment of frames within R. As a first step, movie frames 10–30 (of 40) were each cross-correlated against their higher-index neighbors. Of the pair that yielded the cross correlogram (CCG) with the highest peak value, the frame with the higher index became the first frame of R. In every iteration step, each movie frame was correlated with the sum of all frames in R, excluding the frame itself. The frame with the highest CCG peak among those not yet in R was aligned to the frame sum and incorporated into R. The alignment of any frame already in R was adjusted if the cross-correlation peak amplitude had increased compared to the last iteration. This iterative re-adjustment was continued for at most 100 iterations even after all movie frames had been incorporated into R, until the sum of all cross-correlation peak amplitudes re-occurred. Shift values were determined by 20-fold Fourier padding a $10 \times 10$ pixel region around the cross-correlation peak and taking the location of the maximum. Motion-correction shifts were then applied with sub-pixel resolution as linear phase shifts in Fourier space to the gain-corrected and down-sampled but unfiltered movie frames. Starting from the beginning of the movie, movie frames were then summed until an accumulated electron exposure of 1200 or 1600 electrons/$nm^2$ for apoferritin or DLP images, respectively, was reached. Gold fiducial particles in apoferritin images were masked after these processing steps by substituting all pixel values in those regions of the image with the mean value of the remaining image pixels.

## Template generation

Templates were computed using the structural information in the target's PDB coordinate file retrieved from the RCSB Protein Data Bank (RRID:SCR_012820). Unless stated otherwise, all protein atoms in the file were included using the atom coordinates, structure (B) factors, and occupancy values given in the PDB file. The electrostatic potential map for the target structure was calculated using the TEM-Simulator software package (v. 1.3, ref. [*Rullgard et al., 2011*]), with the potential around the protein set to that of the solvent ($\approx$4.875 V for water). To avoid aliasing artifacts from the sharply peaked atomic potentials we used a voxel pitch of 0.0125 nm for the initial potential calculation, except for the full DLP capsid (PDB:3KZ4) and BSA (PDB:4F5S) where the values were 0.05 nm and 0.025 nm, respectively. We used only the real part of the potential and padded the data with the solvent potential value to fill a cube that was just large enough to contain the structure, subtracted the solvent potential, and resampled the cube (by Fourier cropping) at the sample-referred detector pixel size, $\Delta x$, of the corresponding experimental data. To obtain the scattering potential we multiplied the electrostatic potential by $m_e q_e \lambda_e \Delta x / (2\pi \hbar^2)$, where $q_e$ is the electron charge, and $m_e$ = $1.446 \times 10^{-30}$ kg and $\lambda_e$ = $1.97 \times 10^{-12}$ m are, respectively, the mass and wavelength of the electron at 300 keV. To avoid 'ghosting' artifacts we zero-padded the volume to between two and three times the linear molecule size while keeping the molecule centered.

Projections of the scattering potential were generated using the Fourier projection-slice method (*Levoy, 1992*). The values, $a(\mathbf{k})$, on the slice plane were obtained by cubic-spline interpolation and used to calculate the exit wave ($\psi_{exit} = e^{i\mathcal{F}^{-1}[a(\mathbf{k})]}$) and the wave at the detector: $\psi_{det} = \mathcal{F}^{-1}[\mathcal{F}[\psi_{exit}] \mathrm{CTF}(\mathbf{k})\mathrm{MTF}(\mathbf{k})]$, where $\mathrm{CTF}(\mathbf{k})$ is the contrast transfer function of the microscope and $\mathrm{MTF}(\mathbf{k})$ the modulation transfer function of the detector, which was modeled as described in ref. (*Rullgard et al., 2011*). $\mathcal{F}$ and $\mathcal{F}^{-1}$ denote the Fourier transform operation and its inverse, respectively. Combining MTF and CTF correction improved computational efficiency but

required the assumption that the detector intensity varies approximately linearly with the phase shift. The expectation value of the intensity (rate of electron arrival) at a pixel is given by $|\psi_{det}|^2$ times the average expected rate of electron arrival (taking into account the exposure but not the attenuation by the sample). The contrast transfer function we used was $\mathrm{CTF}(\mathbf{k}) = -sin(\chi(\mathbf{k}))E(\mathbf{k})$, using an aberration function $\chi(\mathbf{k}) = \pi\lambda_e\left(\mathbf{k} \cdot \bar{\bar{f}} \cdot \mathbf{k} - C_s\lambda_e^2|\mathbf{k}|^4/2\right)$ and, to account for the finite illumination coherence (ref. *Reimer and Kohl, 2008*, pp. 230), an envelope function,

$$E(\mathbf{k}) = exp\left(-\left(\frac{\pi\lambda_e|\mathbf{k}|^2 C_c \Delta E}{4V\sqrt{ln(2)}}\right)^2 - \frac{\left(\pi C_s\lambda_e^2|\mathbf{k}|^3 - \pi\bar{\bar{f}}|\mathbf{k}|\right)^2 \alpha_i^2}{ln(2)}\right).$$

$C_s$ and $C_c$ are the spherical and chromatic aberration coefficients, respectively, $\Delta E$ is the energy spread of the source, $\alpha_i$ is the illumination aperture, and $\bar{\bar{f}}$ is a matrix describing defocus and astigmatism, which was estimated from the image using the software package CTFFind4 (*Rohou and Grigorieff, 2015*). For experimental images, $C_s$ and $C_c$ were both assumed to be 2.7 mm, $\alpha_i$ was set to 50 μrad (consistent with what was used by ref. (*Vulović et al., 2013*) after taking into account the different current density), and the MTF parameters were chosen to yield a performance curve similar to what was reported for the very detector we used (*Ruskin et al., 2013*) ($a = 0$, $b = 0.935$, $c = 0$, $\alpha = 0$, $\beta = 0.64$).

## Image simulations

Simulated micrographs were generated using either TEM-Simulator or a custom procedure. When using TEM-Simulator, both phase and amplitude contrast were included and the images were simulated at twice the final resolution and then downsampled to a pixel-size of 0.0965 nm by Fourier cropping. Typical parameters files for TEM-simulator are provided in *Supplementary file 2*.

The custom procedure, which did not account for ice and used only phase contrast, was as follows: for each of the protein species included in the simulation, the scattering potential map was first calculated from the PDB file as described above (see Template generation). For each of the copies of the protein in the virtual sample, the map was rotated to the desired orientation, projected onto the image plane, zero-padded to the edges of the desired image region, and shifted to the specified location in the plane. The sum ($\Sigma_{pp}$) of all the projected protein potentials was then used to calculate the exit wave ($\psi_{exit} = e^{i\Sigma_{pp}}$), from which the expected intensity distribution on the detector was calculated as described above. For the PPM the CTF was set to unity for all pixels with $|\mathbf{k}| \leq 0.1$ cycles nm$^{-1}$ and to the imaginary unit, $i$, everywhere else. The actual pixel values in a simulated image were obtained by multiplying the expected pixel intensity (which varies around one) by the exposure (expressed as electrons/pixel) and then replacing these values with values drawn independently for each pixel from a Poisson distribution with a mean corresponding to the expected electron count for that pixel. Unless noted otherwise, the microscope parameters used for the generation of templates to search a particular simulated image were the same as those used to generate that image. When coherent illumination was assumed, $C_C, \Delta E$, and $\alpha_i$ were all set to zero, and the MTF parameters to the values for perfect transmission.

## Searching the images

CCGs were calculated as follows: we padded the template to the image size with its own mean value, subtracted the mean from the image and the template, and calculated the Fourier transforms, obtaining $I_F$ and $T_F$. When we used a whitening filter in a search, the filter, which comprised the reciprocal square root of the radially averaged power spectral density (PSD) for the image (*Figure 3c*), was calculated and applied to the image and all templates. The low-pass filter used to calculate variable-frequency cutoff curves (*Figure 5a,b*) was applied at this stage. The filtered Fourier-space data and template were normalized so that their real-space distributions had a mean of zero (by setting the pixel at the origin to zero) and a standard deviation of one, yielding $\tilde{I}_F$ and $\tilde{T}_F$. The CCG was then obtained as $\mathcal{F}^{-1}\left[\tilde{I}_F \times \tilde{T}_F^*\right]$, where * denotes the complex conjugate. Throughout, a FFT normalization was used that preserved the quadratic norm, i.e., the sum over the squared absolute values.

A full (orientation) search consisted of calculating CCGs between one experimental or simulated image and a set of 2,359,296 templates, each corresponding to a different orientation of the protein in space. The orientations were generated by an algorithm based on the Hopf fibration (*Yershova et al., 2010*). Orientations were represented by unit quaternions and were converted into rotation matrices where needed. The following results were stored: for each CCG value larger than a given threshold (for most searches equal to 5) the value and the corresponding LOC; for each orientation the maximum CCG value and its location; for each location, the maximum, the mean, and the variance of the distribution of CCG values across all orientations; a histogram of all CCG values. One full search (including the generation of the templates) of a single 1850 × 1850 pixel image required approximately 1000 CPU-hours using compute nodes with 2.7 GHz Intel Sandy Bridge E5-2680 processors and 8 GB of RAM.

## Precision-recall curves

Precision-recall curves were determined by comparing the ground-truth LOCs to locations and orientations in the list of all CCG values larger than five. For DLP images, only those elements in the list were considered that fell within 32.5 nm of the DLP center. Parsing the list in order of descending CCG value, we incorporated an element from that list into the list of true positives whenever the element's LOC fell within both five pixels and three (simulations) or five (experiments) degrees of any ground-truth LOC (see below) but only if no previously examined element had already met those criteria for that particular ground-truth LOC. If the element's LOC was not close to any ground-truth LOC it was incorporated into the list of false positives but only if it was not close to any LOC already in that list, using the same distance criteria. The true positive and false positive counts ($n_{tp}$ and $n_{fp}$) for a particular threshold were then determined by simply counting the elements in the corresponding lists with CCG values above that threshold. Each time an element increased $n_{tp}$, the recall was calculated as the ratio between $n_{tp}$ and the number of ground-truth LOCs and the precision as $n_{tp}$ / ($n_{tp} + n_{fp}$). Precision-recall curves representing multiple datasets (as in *Figure 6c*) were calculated as the mean of all datasets' precision values at a given recall.

To determine the ground-truth LOCs for a particular DLP image, the image was cropped closely (65 pixel margin) and searched using a complete set of ASU templates (PDB: 3KZ4, one copy of chains A-O) generated using our standard set of orientations. For each group of neighboring pixels with at least one CCG MIP value >7.11 the orientation corresponding to the highest-valued CCG in that neighborhood was expanded into a full icosahedrally symmetric set with 60 elements. All such sets of orientations were combined and then clustered using a rotation threshold of three degrees. From the largest cluster we selected the element with the largest CCG value and searched the neighborhood of its orientation in 0.31 degree increments using templates based on the full DLP capsid structure (PDB: 3KZ4) and a whitening filter. The orientation that gave the largest CCG peak was thereafter used as the particle's orientation.

To determine the expected ASU locations in a DLP image, we calculated the 3D coordinates for the center of mass of each ASU reported in the PDB file and rotated them about the particle center to the particle orientation, and projected them onto the image plane. The shift of this pattern relative to the center of the image was calculated as follows: we generated ASU templates at the 60 orientations obtained by expanding the particle's orientation into a full icosahedrally symmetric set. These templates were then whitened and cross-correlated with the whitened image. The resulting CCGs were then combined into a maximum intensity projection (MIP) and cross-correlated with a location reference image. The location of the maximum in that CCG corresponds to the pattern shift vector sought. The location-reference image was created by shifting a discrete-delta-function image (where the pixel at the center was set to one and all other pixels were set to zero) with subpixel resolution to the expected locations of all ASUs whose CCG included a value above 5.47 ($= \sqrt{2}\ \mathrm{erfc}^{-1}(2/(60 \times 850^2))$, see SNR) in the combined MIP and maximum intensity projecting the resulting images.

## Rotavirus VP1 reconstruction

Capsid-bound VP1 molecules were detected as follows. A dataset consisting of 4,178 DLP images (defocus range 300–1900 nm) that had been acquired and pre-processed (including frame-alignment, exposure filtering, and magnification distortion correction as described in *Grant and*

*Grigorieff, 2015*) was used. First, using for each DLP the orientation and defocus from a list generously provided by T. Grant, we performed a search for the 60 ASUs (target structure PDB: 4F5X, one copy of chains A-O only) constrained to the 60 orientations consistent with the DLP orientation provided. Only those images (3,296 of 4,178) where each of the 60 ASUs could be detected within a radius of five pixels (0.5 nm) of its location expected given the orientation of the entire DLP were kept for further analysis, using the coordinates of the CCG peaks instead of the locations expected.

The search for VP1 LOCs was performed by cross-correlating the image with a set of 60 templates generated using only the VP1 fragment of the structure (PDB: 4F5X, one copy of chain W only) with all B-factors set to 100.0 Å$^2$ and at the orientations expected. The locations were constrained to a 1-pixel radius (five pixels per CCG) of the detected ASU locations. CCGs were corrected for variations in mean and variance with the radius from the center of the DLP, which were estimated by searching with control templates (the real templates rotated by 180 degrees around the center of mass in the squared template image), binning the CCG values generated according to the distance of the corresponding pixels from the DLP center, calculating for each bin the mean and standard deviation, and radially smoothing the results using a 5-pixel wide 1$^{st}$-order Savitzky-Golay filter.

The searches for VP1 yielded a set of $2 \times 3296 \times 12 \times 5 \times 5$ CCG values (*Supplementary file 3*), which represent five values (for the predicted pixel and the four closest pixels) for each of the five possible VP1 locations in each of the 12 five-fold vertices in each of the 3,296 particles using two templates (target and control). To produce *Figure 6d*, all potential VP1 locations for which the maximum across the pixel-neighborhood (the last index of *Supplementary file 3*), was above 2.03 were counted as detected VP1s. To produce the VP1 reconstruction (*Figure 6e*) we used a higher threshold (2.56) to reduce the number of expected false alarms to one per DLP, and furthermore counted a VP1 as detected only if no other VP1 was detected at the same vertex. For each detected VP1 in the dataset (15,265 in total), we generated a copy of the original image, padded it to $2048 \times 2048$ pixels, shifted it to position the associated ASU at the center pixel, and together with the expected orientation and defocus, incorporated it into a set that was then provided as input to a development version of Frealign (*Grigorieff, 2007*), which was used to calculate the reconstruction directly, using no refinement, classification, or imposed symmetry.

The 3D difference map comparing the experimental reconstruction to a model was generated in Diffmap (http://grigoriefflab.janelia.org/diffmap). The real part of the electrostatic potential map of the DLP-VP1 complex used in this comparison was generated at 0.05 nm voxel pitch using TEM-simulator (*Rullgard et al., 2011*), for PDB: 4F5X, with 60 copies of chains A-O and one copy of chain W, with all B-factors in chain W set to 100.0 Å$^2$. The map was then rotated to the expected orientation, and aligned with the reconstruction by 3D cross correlation.

To estimate the missed-target rate for a given threshold in the constrained search for VP1, we used the fact that all alarms are false when using the control template. With 11 VP1s per 60 potential locations, there are only 49 locations left for a false alarm to occur. This means that when searching with the real template the average number of false alarms that we need to subtract from the number of detections ($n_d$) to get at the number of true detections ($n_{tp}$) is only 49/60 times as large as that for the control template ($n_{fp}$). The recall rate (one minus the missed target rate, $r_{fn}$) is then the number of true detections divided by the number of target occurrences (11) and

$$r_{fn} = 1 - \left( n_d - n_{fp}\frac{49}{60} \right) / 11.$$

## SNR

How do we know, given a set of image pixel intensities, what the likelihood is that there was an actual target molecule at a particular location? One approach is simply to ask: for which threshold ($\vartheta$) is the likelihood that a CCG pixel value $> \vartheta$ represents an actual target equal to 50%? This occurs when the rate of false alarms becomes equal to the number of targets expected in the image. For one target per full search, with about $10^{13}$ LOCs, the false-alarm rate should be $10^{-13}$ per examined LOC. For Gaussian noise this occurs when $(10^{13} = \mathrm{erfc}\,(\vartheta/(\sigma\sqrt{2}))/2)$, which requires the threshold in the CCG to be about 7.34 times the SD.

The SNR was calculated as the ratio of the CCG value and the standard deviation, $\sigma$, of the CCG noise, which follows the convention adopted by (*Saxton and Frank, 1976*; *Sigworth, 2004*) but

differs from other definitions of SNR also used in the cryo-EM field (*Frank and Al-Ali, 1975*; *Unser et al., 1987*; *Grigorieff, 2000*). For full searches, which yielded large numbers of CCGs (>10$^{12}$), $\sigma$ was estimated by fitting $y = ae^{-(x-x_0)^2/2\sigma^2}$ to the histogram of all CCG values. For full searches of rotavirus DLP images, the resulting SNR values were further normalized by subtracting the mean of the SNR values for all orientations at that location and dividing by their SD.

SNR values in the constrained search for VP1 were corrected as described above (see Rotavirus VP1 reconstruction). To estimate the SNR for the detection of VP1 we determined the SNR that made the following distributions most similar: first, the distribution of the maxima over the last (neighborhood) index of the $2 \times 3296 \times 12 \times 5 \times 5$ CCG matrix that resulted from the restricted search for VP1 (above, also *Supplementary file 3*); second, results from a search that used control templates, modified by adding the SNR value to one randomly selected pixel for one randomly selected potential VP1 location in each of a randomly selected 11-member subset of the 12 vertices in each DLP. The similarity between distributions was assessed by summing the squared differences of mean, variance, and skewness.

## Total number of electrons scattered

To calculate a rough estimate of the total number of electrons scattered by a single apoferritin molecule we added the electron scattering cross-sections (NIST Electron Elastic-Scattering Cross-Section Database: Version 3.2, http://www.nist.gov/srd/nist64.cfm) of all 32760 hydrogen, 20904 carbon, 5832 nitrogen, and 6288 oyxgen atoms at 300 keV, which at an exposure of 1000 electrons-nm$^{-2}$ yields 734.0, 1031.2, 1305.0, and 1420.3 electrons (out of a total of 1435.9 total forward scattered electrons) into aperture half-opening angles of 9.75, 15, 30, and 100 mrad, respectively. 9.75 mrad corresponds to a resolution of 0.2 nm.

Unless stated otherwise, all calculations were implemented using custom scripts written in Matlab R2013a (Mathworks) and executed on individual workstations or on Sandy Bridge nodes of the Janelia Compute Cluster (Intel E5-2680 2.7 GHz, Scientific Linux 6.3). The search code was compiled for single-threaded CPU usage using the Matlab Compiler Toolbox (v. 5.2).

## Code availability

Compiled and source code written in Matlab to calculate reference structures (scattering potential matrices) from Protein Data Bank-formatted models, and to search for these structures in images by cross-correlation, are available for download at https://github.com/jpr-smap/smappoi. git (*Rickgauer, 2017*). A copy is archived at https://github.com/elifesciences-publications/smappoi.

## Acknowledgements

We thank Tim Grant and Alexis Rohou for providing numerous fruitful discussions, Thomas Barends for molecular dynamics calculations used in preliminary simulations, Zhiheng Yu for help with the data acquisition, Stephen Harrison for discussions and for providing the DLP sample, Ruben Diaz-Avalos for help with data acquisition and providing frozen DLP grids, and Cristina Domnisoru, Shawn Mikula, and Alexis Rohou for comments on the manuscript.

## Additional information

### Funding

| Funder | Grant reference number | Author |
| --- | --- | --- |
| Howard Hughes Medical Institute | Internal | J Peter Rickgauer<br>Nikolaus Grigorieff<br>Winfried Denk |

The funders had no role in study design, data collection and interpretation, or the decision to submit the work for publication.

## Author contributions
JPR, Designed and performed experiments, performed most of the simulations, wrote analysis procedures, discussed and interpreted results, and wrote and revised the paper; NG, Designed experiments, discussed and interpreted results, and helped edit the paper; WD, Conceived the project and performed preliminary simulations, designed experiments, wrote analysis procedures, discussed and interpreted results, and wrote and revised the paper

## Author ORCIDs
J Peter Rickgauer, http://orcid.org/0000-0002-6957-4681
Nikolaus Grigorieff, http://orcid.org/0000-0002-1506-909X
Winfried Denk, http://orcid.org/0000-0002-0704-6998

# Additional files

## Supplementary files
• Supplementary file 1. 3D reconstruction. Stack of sections from a reconstruction of the rotavirus RNA polymerase VP1 bound near a fivefold vertex in a double-layered particle (DLP; odd frames), interleaved with sections through a simulated DLP based on the model used for VP1 template generation (even frames). Voxel size is 0.1023 nm.

• Supplementary file 2. Sample input files for TEM-simulator v.1.3 (Rullgard et al., J. Microscopy 243 (3):234–256, 2011) to calculate expected intensity distributions at the detector, and expected output files. Images generated by TEM-simulator and used here were downsampled twofold, to a pixel size of 0.0965 nm, by Fourier cropping before substituting each pixel intensity with a new value drawn from a Poisson distribution with the same mean. Further details of the included files are provided in README.txt in the. zip folder.

• Supplementary file 3. CCG values from the constrained VP1 search. Indices for the five-dimensional matrix ($2 \times 3296 \times 12 \times 5 \times 5$) correspond to: target or control template (1 or 2); DLP image number (1–3296); vertex number (1–12); position within the vertex (1–5); pixel neighborhood (1–5), with #3 the center pixel and 1, 2, 4, and 5 the adjacent pixels.

• Supplementary file 4. Results from full searches described in the main text (see *Supplementary file 5* for file names and descriptions). Each file contains a matrix of the 100,000 highest CCG values in descending order (column 1) and the corresponding LOCs (columns 2–7). Locations in the image (columns 2 and 3, in nm) are given as x and y distances from one corner pixel. Template orientations (columns 4–7) are given as the four sequential elements of a unit quaternion vector q. An equivalent rotation matrix, R, may be calculated from this representation as:

$$R = \begin{bmatrix} q_1^2 + q_2^2 - q_3^2 - q_4^2 & 2(q_2 q_3 - q_1 q_4) & 2(q_2 q_4 + q_1 q_3) \\ 2(q_2 q_3 + q_1 q_4) & q_1^2 - q_2^2 + q_3^2 - q_4^2 & 2(q_3 q_4 - q_1 q_2) \\ 2(q_2 q_4 - q_1 q_3) & 2(q_3 q_4 + q_1 q_2) & q_1^2 - q_2^2 - q_3^2 + q_4^2 \end{bmatrix}$$

• Supplementary file 5. List of files included in *Supplementary file 4*. Columns $\Delta f_1$, $\Delta f_2$, and $\alpha_{ast}$ provide defocus parameters (*Rohou and Grigorieff, 2015*) assumed in template generation.

## Major datasets
The following previously published datasets were used:

| Author(s) | Year | Dataset title | Dataset URL | Database, license, and accessibility information |
|---|---|---|---|---|
| De Val N, Declercq JP | 2008 | Horse spleen apoferritin | http://www.rcsb.org/pdb/explore/explore.do?structureId=2W0O | Publicly available at the RCSB Protein Data Bank (accession no: 2W0O) |

| | | | | |
|---|---|---|---|---|
| Braig K, Otwinowski Z, Hegde R, Bois-vert DC, Joachi-miak A, Horwich AL, Sigler PB | 1995 | THE CRYSTAL STRUCTURE OF THE BACTERIAL CHAPERONIN GROEL AT 2.8 ANGSTROMS | http://www.rcsb.org/pdb/explore/explore.do?structureId=1GRL | Publicly available at the RCSB Protein Data Bank (accession no: 1GRL) |
| Estrozi LF, Settem-bre EC, Goret G, McClain B, Zhang X, Chen JZ., Gri-gorieff N, Harrison SC | 2012 | Location of the dsRNA-dependent polymerase, VP1, in rotavirus particles | http://www.rcsb.org/pdb/explore/explore.do?structureId=4F5X | Publicly available at the RCSB Protein Data Bank (accession no: 4F5X) |
| Lu X, Harrison SC, Tao YJ, Patton JT, Nibert ML | 2008 | Crystal Structure of VP1 apoenzyme of Rotavirus SA11 (N-terminal hexahistidine-tagged) | http://www.rcsb.org/pdb/explore/explore.do?structureId=2R7O | Publicly available at the RCSB Protein Data Bank (accession no: 2R7O) |
| McClain B, Settem-bre EC, Bellamy AR, Harrison SC | 2009 | Crystal Structure of the Rotavirus Double Layered Particle | http://www.rcsb.org/pdb/explore/explore.do?structureId=3KZ4 | Publicly available at the RCSB Protein Data Bank (accession no: 3KZ4) |
| ujacz A, Bujacz G | 2012 | Crystal Structure of Bovine Serum Albumin | http://www.rcsb.org/pdb/explore/explore.do?structureId=4F5S | Publicly available at the RCSB Protein Data Bank (accession no: 4F5S) |

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
