## [Decision Letter]

Thank you for submitting your article "Single-protein detection in crowded molecular environments with high specificity using cryo-EM" for consideration by *eLife*. Your article has been reviewed by two peer reviewers, and the evaluation has been overseen by a Reviewing Editor and Richard Aldrich as the Senior Editor. The following individuals involved in review of your submission have agreed to reveal their identity: Steven J Ludtke (Reviewer #2); Robert M Glaeser (Reviewer #3).

The reviewers have discussed the reviews with one another and the Reviewing Editor has drafted this decision to help you prepare a revised submission.

The method proposed allows one to detect characteristic signatures of macromolecular assemblies in crowded cellular environments without labels. This method is complementary to cryo-EM tomography (cryo-ET) in the sense that the size limit of what can be mapped is much lower than that of cryo-ET, but it appears that sample thicknesses has to be much less than what it can be in tomography. It was felt that this method will be taken up by a large community, and it will make a big impact.

The major claims made for this new method, which are well supported by the data, are:

1) Proteins as small as 150 kDa can be reliably detected in vitreous ice.

2) In a crowded environment, like that of the interior of brain tissue, the molecular weight limit should increase to 300 kDa.

3) Identification of macromolecules is highly specific.

This manuscript demonstrates that it is possible to use very fine-grained template matching to identify component molecules in CryoEM images even when obscured by other material. This study focuses on the unique high resolution features present in individual molecules, rather than the low resolution "shape" information often relied upon in single particle picking software. The method is based on the well-known approach of template matching by cross correlation, in which the templates are known, high-resolution structures of macromolecules whose locations and orientations are being sought. The novel result presented in this manuscript is that such template matching works best when low spatial-frequencies are suppressed, leaving mainly the high-frequency components to dominate the cross-correlation search. It is shown, for example, that the method works best when images are taken so close to focus that one cannot see the particles by eye. This result is highly unexpected, since it is known that the SNR is far better at low frequency than at high frequency. Furthermore, there is well-justified concern in the field that the use of a high-resolution template can produce model-dependent bias. This manuscript properly addresses that concern by setting the threshold for identification very high. This is an interesting study, and worthy of publication in *eLife*.

While it is very nice to see a manuscript on a topic like this really cover the subject in depth and with sufficient technical detail for others to follow in the author's footsteps, the discussion of the various trials and overall discussion seemed at times rather repetitive, covering the same points multiple times. The main manuscript could be considerably streamlined without reducing its scientific content. A bewildering number of points are discussed, and a large number of panels are grouped into just 4 figures. For the sake of clarity, the authors should undertake a major simplification of the manuscript. Even the overall structure of the presentation itself should be reconsidered and improved. For example, it would help if fewer image-panels were used, sufficient to illustrate the concept behind this method and to document that it really works. Further supporting details might then be moved to Supplemental Material.

The method has clearly demonstrated its ability to work on isolated particles and on particles obscured by the bulk of other molecules (the virus substructure example). A key point in this manuscript seems to be the argument that this may be used for molecular localization as an alternative to tomography in cellular material. This study does not make a convincing argument that this will be possible for two reasons:

1) in-situ molecules are likely to undergo substantial structural variations, and will often be complexed with other molecules which will lead to conformational variation. The question of whether the templates being searched for will be sufficiently similar to the actual molecules has not really been addressed. What level of variability would begin to cause failure? This might be addressed very briefly.

2) Their estimate of a 100nm thick layer as being typical for cellular tomograms seems inaccurate. For Cryo studies either very small whole cells are vitrified, which are typically ~500nm thick, or cells are cryo-sectioned or FIB-milled, which are also frequently considerably thicker than 100nm. Perhaps the authors might want to state that while a 100 nm thick sample might be desired for high resolution tomography, this is usually not possible.

For these reasons, it was felt that either the authors need to show some sort of proof of concept on CryoEM of cellular material, or downplay their claims of usefulness for cellular material somewhat. The reviewers are not stating that the method will not work for cells, simply that this manuscript currently provides insufficient evidence to make this claim.

There were some concerns about the use of the term SNR in this manuscript. SNR is widely used in CryoEM for various purposes and the definition here, while somewhat acceptable from a pure image processing perspective, may be confusing to people in CryoEM. Canonically SNR or SSNR are measures of power or intensity ratios. In that "space" coherent averaging scales as amplitude squared and incoherent averaging scales linearly, producing SSNR which scales linearly with "amplitude" whatever that means in a specific situation. In image processing, SNR is sometimes defined as a ratio between a peak and the standard deviation of the background similarly to this manuscript, but this definition produces an amplitude ratio rather than an intensity ratio, giving the SNR here different scaling properties than the SNR and SSNR elsewhere in CryoEM. Very often the SNR in image processing is also squared for this reason. If the authors opt to continue with this definition, it is important that this discrepancy be described in the SNR methods section.

---

## [Author Response]

[…]

While it is very nice to see a manuscript on a topic like this really cover the subject in depth and with sufficient technical detail for others to follow in the author's footsteps, the discussion of the various trials and overall discussion seemed at times rather repetitive, covering the same points multiple times. The main manuscript could be considerably streamlined without reducing its scientific content. A bewildering number of points are discussed, and a large number of panels are grouped into just 4 figures. For the sake of clarity, the authors should undertake a major simplification of the manuscript. Even the overall structure of the presentation itself should be reconsidered and improved. For example, it would help if fewer image-panels were used, sufficient to illustrate the concept behind this method and to document that it really works. Further supporting details might then be moved to Supplemental Material.

One panel was removed and the remaining panels are now instead grouped into 6 figures. We have also streamlined the Discussion by shortening the list of potential applications, and by eliminating a paragraph that restated points made in the Introduction.

Since the Research Article format at *eLife* does not include a separate section for Supplemental Material, our understanding is that text must be either in the manuscript or omitted entirely.

As the reviewers themselves point out, the fact that the method works best for images taken close to focus is surprising. We think that a thorough treatment of the reasons why this is true is beneficial to the manuscript.

The method has clearly demonstrated its ability to work on isolated particles and on particles obscured by the bulk of other molecules (the virus substructure example). A key point in this manuscript seems to be the argument that this may be used for molecular localization as an alternative to tomography in cellular material. This study does not make a convincing argument that this will be possible for two reasons:

1) in-situ molecules are likely to undergo substantial structural variations, and will often be complexed with other molecules which will lead to conformational variation. The question of whether the templates being searched for will be sufficiently similar to the actual molecules has not really been addressed. What level of variability would begin to cause failure? This might be addressed very briefly.

The reviewers are correct to point out that differences between a reference structure and real structures *in situ* should, once sufficiently large, lead to a failure to detect the expected structure. We agree that this topic should be addressed, and have done so as follows:

1) We now include as Figure 2—figure supplement 2 a brief analysis that considers how detectability might be affected by differences between a reference structure and a real structure.

When a part of a protein assumes a different conformation that no longer matches the model, the SNR should be reduced by the ratio between the mass of unchanged part and that of the total protein mass in the original templates. If it is known which parts of the protein are likely to differ from the model, those parts of the model can be removed. In that case the SNR should scale again as the square root of the molecular weight (Figure 2).

2) We now make the following point in the Discussion section: “While the inclusion of higher spatial frequencies in both templates and in the image improves the detection sensitivity (Figure 5), it also exacerbates the differences between a particular reference structure (usually determined in a non-native context such as a crystal) and its conformation *in situ* (Figure 6, [Supplementary-material SD1-data]). Searches may, therefore, need to include template sets for a whole range of possible conformations.”

2) Their estimate of a 100nm thick layer as being typical for cellular tomograms seems inaccurate. For Cryo studies either very small whole cells are vitrified, which are typically ~500nm thick, or cells are cryo-sectioned or FIB-milled, which are also frequently considerably thicker than 100nm. Perhaps the authors might want to state that while a 100 nm thick sample might be desired for high resolution tomography, this is usually not possible.

It is becoming more common to prepare cryo-sections of vitrified cells or tissue samples for tomography that are cut to a thickness of 100 nm or less (see, e.g., reviewer refs. ^1-13^). To clarify that this is the type of application we have in mind, we have modified the text referring to sample thickness in the Discussion section so that it now reads: “Whether detection of proteins above 300 kDa is possible in 100 nm-thick slices of vitrified biological material should be testable with the help of modern cryo-sectioning techniques (Al-Amoudi, Diez, Betts and Frangakis, 2007).”

For these reasons, it was felt that either the authors need to show some sort of proof of concept on CryoEM of cellular material, or downplay their claims of usefulness for cellular material somewhat. The reviewers are not stating that the method will not work for cells, simply that this manuscript currently provides insufficient evidence to make this claim.

The reviewers correctly point out that we have not demonstrated the usefulness of this approach in actual cellular material. To emphasize this, we have modified the manuscript in the following ways:

1) The Abstract now refers to an estimated detection limit in “in 100 nm thick samples of dense biological material” rather than in “100 nm thick tissue sections.”

2) The Introduction now describes an approach “to determine the locations and orientations of proteins in crowded environments” rather than “to determine the intracellular locations…”

3) We have eliminated all text in the Discussion section in which detectability with our approach was compared directly to tomography (which has in fact been tested in cellular material).

There were some concerns about the use of the term SNR in this manuscript. SNR is widely used in CryoEM for various purposes and the definition here, while somewhat acceptable from a pure image processing perspective, may be confusing to people in CryoEM. Canonically SNR or SSNR are measures of power or intensity ratios. In that "space" coherent averaging scales as amplitude squared and incoherent averaging scales linearly, producing SSNR which scales linearly with "amplitude" whatever that means in a specific situation. In image processing, SNR is sometimes defined as a ratio between a peak and the standard deviation of the background similarly to this manuscript, but this definition produces an amplitude ratio rather than an intensity ratio, giving the SNR here different scaling properties than the SNR and SSNR elsewhere in CryoEM. Very often the SNR in image processing is also squared for this reason. If the authors opt to continue with this definition, it is important that this discrepancy be described in the SNR methods section.

We now point out in the Materials and methods section that different definitions of SNR are in use and provide references both for the case where the definition matches ours and for cases where it doesn’t. This section now reads: “The SNR was calculated as the ratio of the CCG value and the standard deviation, *𝜎*, of the CCG noise, which follows the convention adopted by (Saxton and Frank, 1977, Sigworth, 2004) but differs from other definitions of SNR also used in the cryo-EM field (Frank and Al-Ali, 1975; Unser, Trus and Steven, 1987, Grigorieff, 2000).”

1. Al-Amoudi, A.*, et al.* The three-dimensional molecular structure of the desmosomal plaque. *Proc Natl Acad Sci U S A*
**108**, 6480-6485 (2011).

2. Al-Amoudi, A., Diez, D.C., Betts, M.J. & Frangakis, A.S. The molecular architecture of cadherins in native epidermal desmosomes. *Nature*
**450**, 832-837 (2007).

3. Delgado, L., Martínez, G., López-Iglesias, C. & Mercadé, E. Cryo-electron tomography of plunge-frozen whole bacteria and vitreous sections to analyze the recently described bacterial cytoplasmic structure, the Stack. *J Struct Biol*
**189**, 220-229 (2015).

4. den Hollander, L.*, et al.* Skin Lamellar Bodies are not Discrete Vesicles but Part of a Tubuloreticular Network. *Acta Derm Venereol*
**96**, 303-308 (2016).

5. Gruska, M., Medalia, O., Baumeister, W. & Leis, A. Electron tomography of vitreous sections from cultured mammalian cells. *J Struct Biol*
**161**, 384-392 (2008).

6. Gunkel, M.*, et al.* Higher-order architecture of rhodopsin in intact photoreceptors and its implication for phototransduction kinetics. *Structure*
**23**, 628-638 (2015).

7. Hagen, C. & Grünewald, K. Microcarriers for high-pressure freezing and cryosectioning of adherent cells. *J Microsc*
**230**, 288-296 (2008).

8. Hoffmann, C., Leis, A., Niederweis, M., Plitzko, J.M. & Engelhardt, H. Disclosure of the mycobacterial outer membrane: cryo-electron tomography and vitreous sections reveal the lipid bilayer structure. *Proc Natl Acad Sci U S A*
**105**, 3963-3967 (2008).

9. Hsieh, C.E., Leith, A., Mannella, C.A., Frank, J. & Marko, M. Towards high-resolution three-dimensional imaging of native mammalian tissue: electron tomography of frozen-hydrated rat liver sections. *J Struct Biol*
**153**, 1-13 (2006).

10. Hsieh, C.E., Marko, M., Frank, J. & Mannella, C.A. Electron tomographic analysis of frozen-hydrated tissue sections. *J Struct Biol*
**138**, 63-73 (2002).

11. Krähling, A.M.*, et al.* CRIS-a novel cAMP-binding protein controlling spermiogenesis and the development of flagellar bending. *PLoS Genet*
**9**, e1003960 (2013).

12. Masich, S., Ostberg, T., Norlen, L., Shupliakov, O. & Daneholt, B. A procedure to deposit fiducial markers on vitreous cryo-sections for cellular tomography. *J Struct Biol*
**156**, 461-468 (2006).

13. Pierson, J., Ziese, U., Sani, M. & Peters, P.J. Exploring vitreous cryo-section-induced compression at the macromolecular level using electron cryo-tomography; 80S yeast ribosomes appear unaffected. *J Struct Biol*
**173**, 345-349 (2011).